# Neuronal Cannabinoid CB_1_ Receptors Suppress the Growth of Melanoma Brain Metastases by Inhibiting Glutamatergic Signalling

**DOI:** 10.3390/cancers15092439

**Published:** 2023-04-24

**Authors:** Carlos Costas-Insua, Marta Seijo-Vila, Cristina Blázquez, Sandra Blasco-Benito, Francisco Javier Rodríguez-Baena, Giovanni Marsicano, Eduardo Pérez-Gómez, Cristina Sánchez, Berta Sánchez-Laorden, Manuel Guzmán

**Affiliations:** 1Department of Biochemistry and Molecular Biology, Instituto Universitario de Investigación Neuroquímica (IUIN), Complutense University of Madrid, Centro de Investigación Biomédica en Red sobre Enfermedades Neurodegenerativas (CIBERNED), Instituto de Salud Carlos III, Instituto Ramón y Cajal de Investigación Sanitaria (IRYCIS), 28040 Madrid, Spain; cacostas@ucm.es (C.C.-I.); cblazque@ucm.es (C.B.); 2Department of Biochemistry and Molecular Biology, Instituto Universitario de Investigación Neuroquímica (IUIN), Complutense University of Madrid, Instituto de Investigación Hospital 12 de Octubre (i+12), 28040 Madrid, Spain; marseijo@ucm.es (M.S.-V.); s.blasco@ucm.es (S.B.-B.); eduperez@ucm.es (E.P.-G.); cristina.sanchez@quim.ucm.es (C.S.); 3Instituto de Neurociencias, Consejo Superior de Investigaciones Científicas (CSIC) and Universidad Miguel Hernández (UMH), 03550 San Juan de Alicante, Spain; francisco.rodriguezb@umh.es (F.J.R.-B.); berta.lopez@umh.es (B.S.-L.); 4Physiopathologie de la Plasticité Neuronale, NeuroCentre Magendie, U1215 Institut National de la Santé et de la Recherche Médicale (INSERM), Bordeaux Neurocampus, University of Bordeaux, 33077 Bordeaux, France; giovanni.marsicano@inserm.fr

**Keywords:** cannabinoid receptor, endocannabinoid system, melanoma, brain metastasis, glutamate, NMDA receptor, cell proliferation

## Abstract

**Simple Summary:**

An estimated 60% of melanoma patients develop melanoma brain metastases (MBMs). However, the molecular factors that govern the growth of MBMs are still unknown. The excitatory neurotransmitter glutamate has been shown to control the proliferation of various types of cancer cells within the brain parenchyma, but the cellular sources and molecular mechanisms involved in this process remain unclear. By their well-known role in inhibiting synaptic glutamate release, cannabinoid CB_1_ receptors (CB_1_Rs) located on glutamatergic nerve terminals are conceivably well-positioned to control the growth of MBMs. In silico data mining in cancer-genome atlases and in vitro studies with melanoma cell lines supported that a glutamate-NMDA receptor axis drives melanoma cell proliferation. Strikingly, grafting melanoma cells into the brain of mice lacking CB_1_Rs selectively in glutamatergic neurons increased tumour size and concomitantly activated NMDA receptors on tumour cells. Altogether, our findings reveal an unprecedented role of neuronal CB_1_Rs in controlling MBMs.

**Abstract:**

Melanoma is one of the deadliest forms of cancer. Most melanoma deaths are caused by distant metastases in several organs, especially the brain, the so-called melanoma brain metastases (MBMs). However, the precise mechanisms that sustain the growth of MBMs remain elusive. Recently, the excitatory neurotransmitter glutamate has been proposed as a brain-specific, pro-tumorigenic signal for various types of cancers, but how neuronal glutamate shuttling onto metastases is regulated remains unknown. Here, we show that the cannabinoid CB_1_ receptor (CB_1_R), a master regulator of glutamate output from nerve terminals, controls MBM proliferation. First, in silico transcriptomic analysis of cancer-genome atlases indicated an aberrant expression of glutamate receptors in human metastatic melanoma samples. Second, in vitro experiments conducted on three different melanoma cell lines showed that the selective blockade of glutamatergic NMDA receptors, but not AMPA or metabotropic receptors, reduces cell proliferation. Third, in vivo grafting of melanoma cells in the brain of mice selectively devoid of CB_1_Rs in glutamatergic neurons increased tumour cell proliferation in concert with NMDA receptor activation, whereas melanoma cell growth in other tissue locations was not affected. Taken together, our findings demonstrate an unprecedented regulatory role of neuronal CB_1_Rs in the MBM tumour microenvironment.

## 1. Introduction

Cutaneous melanoma, accounting for 1.7% cases of all newly diagnosed primary malignant cancers, is one the most aggressive forms of tumour, determining 0.7% of all cancer deaths [1,2]. The advent of novel clinical options for melanoma treatment, such as BRAF^V600E^, MEK, and immune-checkpoint inhibitors, have greatly contributed to reduce the mortality rate of patients, especially those with advanced unresectable or metastatic melanoma, that nowadays experience long-term disease control [3,4]. Despite these milestone achievements, almost half of these individuals eventually develop melanoma brain metastases (MBMs), a process that causes the death of 60–70% of melanoma patients [3]. Thus, melanoma is a cancer type of leading morbidity and mortality when it comes to brain metastasis [5,6], due, in part, to the limited therapeutical options for its management [7,8], which results in an extremely low survival rate (less than 10% of patients surpassing 3 years). Thus, identifying molecular mechanisms that influence MBM remains an open question of utmost importance to unveil new potential therapeutic targets for this devastating disease.

The endocannabinoid system (ECS; Figure 1) is a widely spread intercellular communication system highly expressed in the brain. It comprises two G protein-coupled receptors (GPCRs), namely type-1 cannabinoid receptor (CB_1_R) and type-2 cannabinoid receptor (CB_2_R); their endogenous ligands (so-called “endocannabinoids”), namely the eicosanoid lipids *N*-arachidonoylethanolamine (anandamide, AEA) and 2-arachidonoylglycerol (2-AG); and the enzymes involved in endocannabinoid synthesis (mostly *N*-acyl-phosphatidylethanolamine phospholipase D (NAPE-PLD) for AEA, and diacylglycerol lipase α/β (DAGLα/β) for 2-AG) and degradation (mostly fatty acid amide hydrolase (FAAH) for AEA, and monoacylglycerol lipase (MAGL) for 2-AG) [9].

Substantial preclinical research conducted on mouse models has implicated the ECS in the control of the growth and spreading of many types of cancer cells, including, among others, glioblastoma, lung carcinoma, skin carcinoma, hepatocellular carcinoma, breast carcinoma, prostate carcinoma, pancreatic carcinoma, colorectal carcinoma, head and neck carcinoma, cholangiocarcinoma, thyroid epithelioma, rhabdomyosarcoma, lymphoma, and, most relevant to the present study, melanoma cells [10,11,12]. Nonetheless, the great majority of these studies have aimed to pharmacologically manipulate the ECS expressed by the tumour cell, letting aside the ECS in non-tumour cells, thus neglecting a likely important site of physio-pathological action of this system. Of note, CB_1_R is one of the most abundant GPCRs in the mouse and human brain, whereas CB_2_R is scarcely expressed in this organ, being largely restricted to microglial cells [13,14]. The best-established neurobiological action of CB_1_R is the retrograde inhibition of synaptic activity by reducing presynaptic neurotransmitter release. This occurs in numerous neuronal populations, including (inhibitory) GABAergic neurons and (excitatory) glutamatergic neurons, thereby making CB_1_R a key homeostatic signalling platform for brain function [15].

Recent evidence highlights the importance of brain-residing cells for the formation, development, and progression of brain metastases from several types of cancer [16]. Specifically, MBM cells, through secretion of amyloid beta, induce an anti-inflammatory, pro-metastatic phenotype in astrocytes [17,18], and other cancer cells, such as those from breast and lung carcinomas, utilise gap-junctions to enforce astrocytes into secreting pro-tumorigenic signals [19]. Less is known about the role of neurons in the metastatic-cell microenvironment. Recently, it was found that metastatic breast cancer cells establish synapse-like contacts with neurons to co-opt neuron-secreted glutamate. This glutamate engages *N*-methyl-D-aspartate receptors (NMDARs) on the target cancer cell, thereby fuelling an NMDAR-driven oncogenic signalling axis that ensures tumour growth [20]. A similar process occurs in glioma cells, where glutamatergic neuronal activity promotes their proliferation [21,22,23]. However, the mechanisms that govern the output of pro-tumorigenic glutamate from neurons to cancer cells, including MBM cells, are currently unknown.

Here, upon analysing in silico the expression changes of ECS elements from primary melanoma to metastatic melanoma samples, we noticed that profound changes in the expression profile of glutamate receptors occur as well. This prompted us to conduct mechanistic studies on melanoma cells in vitro, which revealed that NMDARs drive melanoma cell proliferation, suggesting that glutamate could boost MBM spreading. As it is well established that CB_1_Rs inhibit glutamate secretion from nerve terminals, we carried out various melanoma cell-based allografting approaches in genetically engineered, immunocompetent mice devoid of CB_1_Rs selectively in either glutamatergic or GABAergic neurons. These experiments showed that CB_1_Rs located on glutamatergic neurons, but not on GABAergic neurons, play an important role in restraining MBM cell growth in vivo. In sum, this study unveils an unprecedented molecular mechanism to explain how glutamate release from neurons is controlled to sustain brain cancer cell growth.

## 2. Materials and Methods

### 2.1. Interrogation of GDC-TCGA Melanoma Datasets

We accessed data from the GDC-TCGA SKCM using Xena [24]. The mRNA expression of selected genes, calculated as fragments read per kilobase million (FPKM) through high-throughput sequencing, was obtained for all samples, and then plotted as primary tumour or metastatic, according to the corresponding sample source upon filtering the patients’ accession numbers by the key words “primary tumour” or “metastatic”. Finally, primary tumour samples were set as 100%, metastatic samples were calculated proportionally, and comparisons were made by an unpaired Student’s *t* test with Welch’s correction.

Survival curves for *GRIN3A* expression were calculated with the same dataset. Groups were divided into quartiles, and the 25% of samples with higher expression were defined as the high *GRIN3A* group, whereas the rest were classified as low *GRIN3A*. Comparisons were made by a log-rank (Mantel–Cox) test.

### 2.2. Cells

The B16.F10 and YUMM1.7 melanoma cell lines were obtained from ATCC (CRL-6475 and CRL-3362, respectively) and maintained in DMEM (Sigma-Aldrich, St. Louis, MO, USA, #D5796) supplemented with 10% FBS and 1% penicillin-streptomycin. The 1014 melanoma cell line was kindly provided by Dr. Lionel Larue (Institut Curie, Paris, France) [25] and maintained in F12 media (Sigma-Aldrich, #N6658) supplemented with 10% FBS and penicillin-streptomycin. Cells were tested for mycoplasma contamination every two weeks by PCR.

### 2.3. Quantitative PCR

RNA was isolated with the NucleoZOL one-phase RNA purification kit (Macherey-Nagel #740404.200) following the manufacturer’s instructions. An amount of 2 µg of total RNA was retro-transcribed using the Transcriptor First Strand cDNA Synthesis Kit (Roche Life Science, Penzberg, Upper Bavaria, Germany, #04379012001) with random hexamer primers. Real-time quantitative RT-PCR (Q-PCR) was performed in a QuantStudio 7/12k Flex System (Applied Biosystems) using the LightCycler^®^ Multiplex DNA Master (Roche Life Science #07339577001) and SYBR green (Roche Life Science #4913914001). The primers used are listed in Table 1.

### 2.4. Cell Viability Assays

For all three cell lines, 6000 cells per well were seeded on a 48-well plate in 10% FBS-containing media. The next day, serial dilutions of MK-801 (Sigma-Aldrich, #M107 dissolved in DMSO), NBQX (Tocris, #1044 Bristol, UK, dissolved in water), and LY341495 (Tocris, #1209 dissolved in DMSO) were prepared and directly added at the indicated concentrations in triplicate. The amount of vehicle was constant between wells, and triplicate incubations were also run with vehicle alone. Forty-eight hours later, 3-(4,5-dimethylthiazol-2-yl)-2,5-diphenyltetrazolium bromide (MTT) (Panreac AppliChem, Chicago, IL, USA, #A2231) was added for 4 h, the media was subsequently aspirated, and the OD at 470 nm was measured upon addition of 100 µL of acid isopropanol per well using a Rayto RT-6100 Microplate Reader. Viability was calculated as the mean of three to five independent experiments relative to the vehicle condition, and statistical comparisons were made by a one-way ANOVA with Dunnett’s post-hoc test.

### 2.5. Immunofluorescence

For all three cell lines, 20,000 cells per well were seeded on coverslips previously coated with poly-L-lysine (10 µg/mL, Sigma-Aldrich, #P9155 at 10 µg/mL concentration) and placed in a 24-well plate. The next day, MK-801 (dissolved in DMSO) or its corresponding vehicle was added at 0.4 mM final concentration. Twenty-four hours later, cells were washed twice with PBS, fixed with 10% formalin (Panreac AppliChem, #143091.1214) at room temperature (RT) for 10 min, again washed twice with PBS, and kept refrigerated.

To immunodetect Ki67, antigens were first retrieved by incubating the samples with citrate buffer (pH 6.0) for 20 min at 65 °C, and then permeabilized and blocked in PBS containing 0.25% Triton X-100 (PBS-TX) and 10% goat serum (Abcam, Cambridge, UK, #ab7481) for 1 h at RT. An antibody against Ki67 (BD Pharmingen, San Diego, CA, USA, #550609) was diluted (1:500) directly into the blocking buffer and incubated overnight at 4 °C. After 3 washes with PBS-TX for 10 min, samples were subsequently incubated for 1.5 h at RT with the appropriate highly cross-adsorbed anti-mouse AlexaFluor 488 secondary antibody (1:500; Thermo Scientific, Waltham, MA, USA #A-11001), together with DAPI (Roche, Basel, Switzerland, #10236276001) to visualize nuclei. After washing 3 times in PBS, sections were mounted onto microscope slides using Mowiol^®^ mounting media. Samples were analysed with a Leica SP2 confocal microscope (Leica Microsystems, Wetzlar, Gemany) and processed with ImageJ software (NIH, Bethesda, WA, USA). In all cases, three biological replicates, each composed of six independent fields, were quantified. Data are presented as the percentage of cells positive for Ki67 divided by the total number of cells in the field, and statistical comparisons were made by an unpaired Student’s *t* test. Representative images for each condition were prepared for figure presentation by applying brightness and contrast adjustments uniformly using ImageJ.

For tumour samples (see below), mice were perfused transcardially with PBS followed by 4% paraformaldehyde solution. Brains were dissected and post-fixed overnight in the same solution, cryoprotected with sucrose, and mounted on standard cryomold with OCT compound. Serial coronal sections (30 μm-thick) through the whole tumour were collected in cryoprotective solution as free-floating sections and stored at −20 °C. Slices were permeabilized and blocked in PBS containing 0.25% Triton X-100 and 10% goat serum (Pierce Biotechnology) for 1 h at RT. For PCNA detection, antigens were retrieved by incubation with citrate buffer (pH 6.0) at 95 °C for 20 min prior to the blocking step. Primary antibodies were diluted directly into the blocking buffer and samples were incubated overnight at 4 °C with the following dilutions: anti-PCNA (1:250, Abcam, #ab-29), anti-PSD-95 (1:250, Abcam, #ab18258). After 3 washes with PBS-TX for 10 min each, samples were subsequently incubated for 1.5 h at RT with the appropriate highly cross-adsorbed anti-mouse, -guinea pig, or -rabbit AlexaFluor 488 and Alexa Fluor 594 secondary antibodies (1:500, all from Invitrogen) together with DAPI (Roche, Basel, Switzerland) to visualize nuclei. After washing 3 times in PBS, sections were mounted onto microscope slides using Mowiol^®^ mounting media. Confocal fluorescence images were acquired by using LAS-X software with an SP8 confocal microscope (Leica Microsystems, Mannheim, Germany). All quantifications were obtained from a minimum of 3 sections per animal, and a minimum of 6 animals per group (as indicated in the corresponding figure legend) was included. Images were taken using apochromatic oil-immersion 40X objective, and standard (1 Airy disc) pinhole. Immunoreactive area was measured using FIJI ImageJ open-source software, establishing a threshold to measure only the specific signal that was kept constant along the different images. Controls were included to ensure that none of the secondary antibodies produced any significant signal in preparations incubated in the absence of the corresponding primary antibodies. Representative images for each condition were prepared for figure presentation by applying brightness, contrast, and other adjustments uniformly.

### 2.6. Animals

Experimental procedures were performed in accordance with the guidelines and approval of the Animal Welfare Committees of Universidad Complutense de Madrid and Comunidad de Madrid, the CSIC, and the Generalitat Valenciana, following the directives of the Spanish Government and the European Commission. Animal housing, handling, and assignment to the different experimental groups was conducted as described previously [26]. Adequate measures were taken to minimise pain and discomfort of the animals. We used conditional CB_1_R*^floxed/floxed;Nex1-Cre^* (herein referred to as Glu-CB_1_R-KO) knockout mice, in which the CB_1_R-encoding gene (*Cnr1*) has been selectively deleted from glutamatergic neurons of the dorsal telencephalon *sensu lato* (including neocortex, paleocortex, and archicortex), and conditional CB_1_R*^floxed/floxed;Dlx5/6-Cre^* (herein referred to as GABA-CB_1_R-KO) mice, in which *Cnr1* has been selectively deleted from GABAergic neurons of the forebrain [27,28]. *Cnr1*-floxed littermates devoid of Cre recombinase were used as controls. We employed animals of both sexes (at an approximate 1:1 ratio), differentially represented in the respective dot plots, and discarded sex-specific effects by independent statistical analysis.

### 2.7. Tumour Allografts

For subcutaneous allografts, 2 million 1014 cells resuspended in 100 µL of PBS were subcutaneously injected in the right flank of adult (2–4-month-old) Glu-CB_1_R-KO and control littermates of both sexes. Tumours were measured every other day with an external calliper, and volume was calculated as 0.52 × (width)^2^ × (length). Animals were euthanised when the tumour volume exceeded 1000 mm^3^. Statistical comparisons were made by an unpaired Student’s *t* test at each timepoint.

For intracranial allografts, adult (2–4-month-old) Glu-CB_1_R-KO, GABA-CB_1_R-KO, and control littermates of both sexes were anaesthetized with 4% isoflurane (Solvet, L’Hospital de Llobregat, Barcelona, Spain, #ESPT0001), subsequently treated with a mixture of buprenorphine (0.1 mg/kg) and meloxicam (1 mg/kg), and placed into a stereotaxic apparatus (World Precision Instruments, Sarasota, FL, USA). Mice were injected with 25,000 1014 cells resuspended in 2 µL of PBS with a Hamilton microsyringe (Sigma-Aldrich #HAM7635-01) coupled to a 30 g-needle controlled by a pump (World Precision Instruments, #SYS-Micro4) directly in the right dorsal striatum (2 µL at a rate of 1 µL/min) with the following coordinates (in mm from bregma): anterior–posterior: +1.00 mm, dorsal–ventral: −2.50 mm, medial–lateral: +2.00 mm. Following each injection, the syringe remained positioned for 5 min before removal. After surgery, mice were monitored daily, and those animals failing to recover pre-surgery body weight were euthanised and not included in the experiment.

### 2.8. Magnetic Resonance Imaging (MRI)

Magnetic resonance imaging (MRI) studies were performed at BioImaC (ICTS BioImagen Complutense), node of the ICTS ReDIB (https://www.redib.net/, accessed on 18 April 2023), using 1 Tesla benchtop MRI scanner (ICON 1T-MRI; Bruker BioSpin GmbH, Ettlingen, Germany). The system consists of a 1T permanent magnet with a gradient system capable of supplying 450 mT/m gradient strength. A solenoid mouse head RF-coil was employed. Animals were anesthetised using 2% isofluorane (IsoFlo, Zoetis, NJ, USA). A longitudinal MRI study to assess and quantify tumour evolution was performed at 12 days after tumour cell injection. Post-contrast images were acquired at the same time (~15 min) after intraperitoneal injection of 0.1 mL of Gd-BOPTA (MultiHance^®^, Gd-BOPTA; Bracco Imaging SpA, Milano, Italy). Routine pre- and post-contrast (MultiHance^®^) MRI studies were acquired to ensure contrast changes due to the gadolinium-based contrast agent.

MRI data were acquired using the software package Paravision 6.0.1 (Bruker, BioSpin). The main MRI protocol consisted of a two-dimensional T1 weighted experiment (T1WI) and a multiecho proton density (PDWI)/T2 weighted (T2WI) experiment. T1WI coronal anatomical sections were acquired using a spin echo sequence with a repetition time (TR) = 267 ms, echo time (TE) = 6.15 ms, number of averages (NA) = 30, field of view (FOV) = 17 × 17 mm^2^, slice thickness = 1 mm, and number of slices = 15. The acquired matrix size was 170 × 170 (resolution 0.100 × 0.100 × 1.00 mm) and the total acquisition time was ~11 min. The PDWI/T2WI experiment was acquired with the same anatomic orientation using a spin multiecho sequence with a TR= 2250 ms, TE = 32 and 96 ms, NA = 4, FOV = 17 × 17 mm^2^, slice thickness = 1 mm, and number of slices = 15. The acquired matrix size was 128 × 128 (resolution 0.132 × 0.132 × 1.00 mm) and the total acquisition time ~3.5 min.

MRI data were analysed using ImageJ software package (Rasband W., National Institutes of Health, Bethesda, MD, USA, version 1.51a). Tumour volume was calculated using the Image J software from T1-weighted images, and statistical comparisons were made by an unpaired Student’s *t* test.

### 2.9. Western Blotting

Mice bearing intracranial tumours were sacrificed 12 days after tumour cell implantation, and their brains were dissected. Melanin-containing (black-coloured) tumours were dissected under a magnifying glass. Then, tumour samples were homogenised independently with the aid of an automated grinder (DWK Life Sciences GmbH, Mainz, Germany, #749540-0000). Almost the entire tumour (~30 µg total protein) was resolved using PAGE-SDS followed by transfer to PVDF membranes with Bio-Rad FastCast^®^ reagents and guidelines. Membranes were blocked with 5% BSA (*w*/*v*) in TBS-Tween-20 (0.1%) for 1 h and incubated overnight with the following antibodies and dilutions: anti-phospho-NMDAR2B-Y1252 (1:1000, Thermo Fisher, #48-5200) and anti-α-tubulin (1:10,000, Sigma-Aldrich #T9026). Both antibodies were prepared in TBS Tween-20 (0.1%) with 5% BSA (*w*/*v*). Membranes were then washed three times with TBS-Tween-20 (0.1%), and HRP-labelled secondary antibodies, selected according to the species of origin of the primary antibodies (Sigma-Aldrich #NA-931 and #NA-934), were added for 1 h at a 1:5000 dilution in TBS-Tween-20 (0.1%) at RT. Finally, protein bands were detected by incubation with an enhanced chemiluminescence reagent (Bio-Rad, Hercules, CA, USA). All results provided represent the densitometric analysis, performed with Image Lab software (Bio-Rad, Hercules, CA, USA), of the phospho-NMDAR2B-Y1252-band optical density vs. the α-tubulin-band optical density, and statistical comparisons were made by an unpaired Student’s *t* test. Uncropped scans of all blots are shown in the Appendix A.

### 2.10. Intracarotid Artery Injection

Mice were anesthetised with isoflurane, injected with buprenorphine (0.1 mg/kg), and laid on a heated glass surface under a stereomicroscope. A ventral cut was performed in the left side of the neck, and skin and fat were retracted to expose the trachea and pectoral muscle where the carotid artery lays. The artery was isolated with thin forceps, detaching it from the bottom of the tissue and from the vagus nerve, which was left untouched. Two sutures were placed under the artery with open knots. Then, the upper external branch of the artery was isolated and firmly closed with suture and a piece of wet cotton placed under the common carotid artery. The bottom suture was subsequently closed, and cells were injected (0.2 million 1014 cells in 100 µL of sterile PBS) in the common carotid artery. The upper suture was closed, the cotton was removed, and the wound was cleaned and closed with surgical stapples. Finally, animals were injected with 0.5–1.0 mL of saline subcutaneously to prevent dehydration in the following hours post-surgery. The animals were monitored in the following hours for proper recovery. A week later, staples were removed.

### 2.11. Brain Metastatic Burden Analysis

Mice injected with tumour cells in the carotid artery were sacrificed and brains were collected and fixed in 4% PFA overnight. This was followed by 3 washes with PBS and incubation in increasing concentrations of sucrose until 30% *v*/*v* with ddH_2_O. Then, brains were cut using a cryotome to obtain 100-µm-thick slices, preserved in freezing media (30% ethylene glycol, 30% ddH_2_O, 30% glycerol, and 10% 10X PBS), and kept at −20 °C. Slices were mounted on glass slides with Mowiol. Slices were imaged using the Axioscan 7 (Zeiss) with brightfield imaging using a 10X objective. The images were exported in high-quality Tiff format, and total brain area and total metastatic area were quantified using ImageJ software. Statistical comparisons were made by an unpaired Student’s *t* test.

### 2.12. Tail Vein Injection

Adult (2–4-month-old) Glu-CB_1_R-KO and control littermates of both sexes were introduced for 5 min in a Thermacage apparatus (Thermo Fisher, #NC1727075) set at 37 °C before surgery. Then, animals were immobilized in a rodent restrainer (Panlab, Barcelona, Spain #LE5016), and 0.5 million 1014 cells in 100 µL of sterile PBS were injected in the lateral tail vein using a 30 g needle. Three weeks later, the mice were sacrificed and metastatic black *foci* in the lungs were counted manually under a magnifying glass. Representative images were taken with a phone camera after fixing and dehydrating the lungs with 4% paraformaldehyde and 70% ethanol, respectively. Statistical comparisons were made by an individual unpaired Student’s *t* test.

## 3. Results

### 3.1. Dysregulation of ECS Elements and Glutamate Receptors in Melanoma Metastases

The expression of ECS elements is dysregulated in multiple forms of cancer, including melanoma [29,30]. To assess if the ECS influences the development of melanoma metastases, we interrogated the GDC-TCGA skin cutaneous melanoma (SKCM) transcriptomic dataset for the expression of the most characteristic ECS elements in primary tumour vs. metastatic biopsies using the Xena^®^ tool [24]. CB_1_R (encoded by *CNR1*) and mostly CB_2_R (*CNR2*), as well as DAGLα (*DAGLA*), were upregulated in metastatic samples, whereas DAGLβ (*DAGLB*), FAAH (*FAAH*), and alpha/beta-hydrolase domain-containing 6 and 12 (*ABHD6* and *ABHD12*)-two alternative enzymes for 2-AG inactivation- were downregulated (Figure 1). All other ECS members analysed remained unaltered (Figure 1). In full caption, these changes, although modest in magnitude, would point to an elevated 2-AG/AEA tone in metastatic melanoma owing to an increased 2-AG production (upregulation of DAGLα) and a reduced 2-AG/AEA bioconversion (downregulation of ABHD6/12 and FAAH, respectively), in concert with an elevated availability of the two bona fide 2-AG/AEA molecular targets, i.e., CB_1_R and CB_2_R.

Of note, endocannabinoid production in the brain occurs upon Ca^2+^ mobilization by G_q/11_ protein-coupled GPCRs, especially metabotropic glutamate receptors 1 and 5 (mGluR1/5) [9]. In addition, an aberrant glutamatergic signalling has been previously found in melanoma [31,32] and other types of cancer [33,34]. Hence, we next analysed the status of glutamate receptors. The expression of mGluR1 (*GRM1*) and mGluR5 (*GRM5*) was conserved from primary tumours to metastatic samples, but moderate changes were evident for other receptors such as mGluR2/3/7/8 (*GRM2, GRM3, GRM7*, and *GRM8*) (Figure 2A). Regarding the subunits forming ionotropic glutamate receptors, namely *N*-methyl-D-aspartate receptors (NMDARs) and alpha-amino-3-hydroxy-5-methyl-4-isoxazolepropionic acid receptors (AMPARs), expression changes were also evident between primary and metastatic samples (Figure 2B,C). Noteworthy, among NMDAR subunits, an upregulation of NMDAR2B (*GRIN2B*) in metastases was the most pronounced change (Figure 2B), whereas the overexpression of GLUR2 (*GRIA2*) could be highlighted for AMPAR subunits (Figure 2C). Changes were also noticeable for PSD-95 (*DLG4*), but not for GKAP (*DLGAP1*) or BDNF (*BDNF*), three proteins downstream of NMDARs (Figure 2D) [20]. Taken together, these transcriptomic analyses unveil a dysregulation (conceivably an overactivation) of glutamatoceptive signalling in metastatic melanoma compared to primary melanoma.

### 3.2. Blockade of NMDARs Inhibits Melanoma Cell Proliferation In Vitro

We next assessed whether melanoma cells express glutamate receptors and, if so, whether these receptors are important to sustain cell growth. For this purpose, we employed three different melanoma cell lines syngenically derived from the C57BL6 mouse strain: the YUMM1.7 cell line [35], the 1014 cell line [25], and the B16.F10 cell line [10]. These cell lines broadly account for two major melanoma-driving genetic mutations that are usually exclusive both in cell lines and tumours and are present in 60–80% of melanomas [36], namely BRAF^V600E/K^ (YUMM1.7) and NRAS^Q61R/K/L^ (1014), whereas B16.F10 contains wild-type forms of both proteins [37]. Quantitative PCR experiments showed that the three cell lines tested express various mGluR subtypes, AMPAR and NMDAR subunits, as well as NMDAR-related proteins, such as PSD-95 (*DLG4*) and GKAP (*DLGAP1*) (Table 2).

As an initial approach to ascertain the importance of glutamatergic signalling in melanoma cell growth, we conducted cell viability assays in the presence of (i) LY341495, an mGluR *pan*-antagonist [38]; (ii) NBQX, an AMPAR-selective antagonist, [39]; or (iii) MK-801, an NMDAR-selective antagonist [40]. Blockade of mGluRs for 48 h only reduced YUMM1.7 and B16.F10 cell viability at the highest dose tested (Figure 3A). We found a similar effect when inhibiting AMPARs, with solely the highest dose slightly decreasing the viability of the 1014 cell line (Figure 3B). In contrast, pharmacological blockade of NMDARs markedly and dose-dependently reduced cell viability in all cases (Figure 3C). To further support this effect, we treated cells with a submaximal dose of MK-801 (0.4 mM) for 24 h, and immunodetected the cell-proliferation marker, Ki67. The percentage of Ki67^+^ cells significantly decreased upon MK-801 challenge (Figure 3D). Taken together, these data support that NMDARs promote melanoma cell proliferation.

### 3.3. CB_1_Rs Located on Glutamatergic Neurons Control the Growth of MBMs In Vivo

Given the well-characterised role of CB_1_Rs in the control of glutamate release by nerve terminals [15] and the aforementioned data showing that NMDARs drive melanoma cell growth in vitro, we reasoned that neuronal CB_1_Rs could limit melanoma cell growth in the brain via a reduction of glutamate release. To assess this idea, we injected 1014 cells into the striatum of conditional knockout mice lacking the CB_1_R gene selectively in cortical glutamatergic/excitatory neurons (CB_1_R*^floxed/floxed;Nex1-Cre^* mice, herein referred to as Glu-CB_1_R-KO mice), and we analysed tumour growth by magnetic resonance imaging (MRI). These animals have increased glutamate levels in the striatum owing to the aforementioned CB_1_R genetic deletion, which leads to an enhanced glutamate release from excitatory cortical neurons that project onto the striatum [27,41]. Twelve days after implantation, tumours were clearly visible, and their volume was almost double in Glu-CB_1_R-KO mice than in wild-type control littermates (Figure 4A). In bare contrast, altering the release of the inhibitory neurotransmitter GABA upon deletion of the CB_1_R gene selectively from forebrain GABAergic/inhibitory neurons [27] (CB_1_R*^floxed/floxed;Dlx5/6-Cre^* mice, herein referred to as GABA-CB_1_R-KO mice) did not cause any overt alteration in tumour growth when using the same experimental setting (Figure 4A). Accordingly, the expression of the cell proliferation marker PCNA was increased in tumours grafted in Glu-CB_1_R-KO animals compared to control littermates, whereas this effect was not observed in GABA-CB_1_R-KO animals (Figure 4B).

When assessing the status on NMDAR-associated signalling in tumour cells, we found that the expression of PSD-95, a protein essential to ensure NMDAR localization on surface membranes and post-synapses [42], was also selectively increased in tumours from Glu-CB_1_R-KO animals, thus supporting that a glutamate-NMDAR signalling axis fosters melanoma cell proliferation (Figure 5A). Immunodetection of NMDAR2B phosphorylated at Y1252 (p-NMDAR2B) has been previously used as a proxy for NMDAR activation [20,43]. Unfortunately, despite a number of attempts, we were unable to detect a reliable staining of p-NMDAR2B by immunofluorescence procedures in our samples using commercially available antibodies, and the high abundance of melanin precluded an immunohistochemical approach as previously employed by others [20,43]. Thus, as an alternative, we obtained tumour extracts from a different cohort of animals and blotted them for p-NMDAR2B. Of note, a marked increase in p-NMDAR2B levels was evident in tumour samples from Glu-CB_1_R-KO mice vs. their corresponding controls, whereas this effect was not evident in tumours from GABA-CB_1_R-KO mice (Figure 5B).

Next, we aimed to analyse whether the action of neuronal CB_1_Rs in controlling glutamate release was also important for cell extravasation during MBM formation. To study the initial colonisation steps of MBMs rather than only cell proliferation [44], we injected 1014 cells in the carotid artery of Glu-CB_1_R-KO and GABA-CB_1_R-KO mice, and analysed tumour burden in these animals (Figure 6A). The data revealed no significant differences in mutant mice as compared to their respective wild-type control littermates (Figure 6A), suggesting that the increased tumour volume observed in the intracranial-allograft approach is mostly due to an increased cell proliferation and is not related to an altered cell extravasation. Likewise, tail vein-injection [20] of 1014 cells colonized the lungs similarly in Glu-CB_1_R-KO than in control animals (Figure 6B). To further evaluate the tissue specificity of this brain CB_1_R/glutamate/NMDAR proliferative axis, 1014 cells were injected subcutaneously in the flank of Glu-CB_1_R-KO mice and their control littermates. No differences in tumour onset or tumour growth were evident between the two genotypes (Figure 6C). Taken together, these experiments support a pro-tumorigenic environment specific to the brain of Glu-CB_1_R-KO mice, most likely due to a disinhibition of glutamate release from excitatory nerve terminals.

## 4. Discussion

Despite milestone achievements in the treatment of melanoma, many individuals die from this devastating disease, particularly those who develop MBMs. Unfortunately, only a few and hardly effective therapeutic interventions exist for MBMs [3]. Thus, the identification of molecular mechanisms that influence the growth of melanoma cells selectively in the brain is crucial for identifying novel pharmacological targets. Here, combining in silico data mining with in vitro and in vivo experimental approaches, we unveil an unprecedented signalling axis for MBM cell proliferation that involves CB_1_Rs, neuron-secreted glutamate, and the activation of melanoma cell NMDARs.

This pro-tumorigenic action of glutamate adds to previous observations reporting that this neurotransmitter influences the proliferation of glioma cells and promotes the invasive tumour growth of breast, pancreatic, and neuroendocrine cancers through AMPARs and NMDARs [20,21,22,34]. In fact, similar effects occur in melanoma cells, which thrive when exposed to high glutamate levels [45], and eventually die upon exposure to the glutamate-release inhibitor riluzole [46]. In line with previous reports, we show that melanoma cells express several glutamate receptors, including NMDARs [47,48], and that their pharmacological blockade (with MK-801) impairs melanoma cell proliferation. These observations resemble those obtained with other cancer-cell types [49], as well as those with a melanoma-cell-based xenograft model [47]. Nonetheless, despite this plethora of evidence, the role of glutamate as a brain-specific proliferative factor for melanoma had not been addressed in detail yet.

So far, most studies had considered glutamate as an autocrine signal produced by the cancer cell itself [50,51], including the melanoma cell [32,52] and, to the best of our knowledge, the potential neuronal origin of this pro-oncogenic glutamate upon MBMs has been overlooked. We consider this latter possibility plausible for several reasons: (i) other metabolites abundant in the central nervous system, such as lactate and ketone bodies, promote melanoma metastasis [53]; (ii) melanoma and other cancer cells establish in the brain ways of communication with neurons and astrocytes, thereby hijacking neuronal and glial signalling pathways [17,18,20,21,22,23]; and (iii) melanoma cells adopt in the brain a neuron-like, brain-adaptive phenotype that upregulates several genes involved in synapse formation (e.g., SNCA), cell adhesion (e.g., LRRC1), and the sensing of neurotrophic factors (e.g., NGFR) [54,55], all of which strongly supports an interaction with brain components. Interestingly, a recent report showed that treating mice with glutamate scavengers reduces blood and cerebrospinal fluid glutamate concentrations in association with a decrease in brain melanoma cell growth [56]. Thus, glutamate seems an important pro-tumorigenic factor for brain melanoma cells and, logically, glutamatergic neurons represent a potential glutamate source. Following this notion, we found that 1014 melanoma cells injected in the brain (but not when inoculated in other sites) of Glu-CB_1_R-KO mice proliferate more markedly than in wild-type or GABA-CB_1_R-KO mice. The intracranial-injection model is ideal to identify factors that foster brain melanoma cell growth, which represents a critical step for the clinical management of MBMs rather than cell invasion/extravasation, as most patients already display metastatic foci when diagnosed [57]. Tumours growing in the brain of Glu-CB_1_R-KO mice exhibited a higher activation of NMDARs (as assessed by phosphorylation of the NMDAR2B subunit and the expression of the NMDAR-adaptor protein PSD-95) and an increased expression of the proliferation marker PCNA. This observation strongly supports the existence of a neuron-melanoma cell crosstalk, given that glutamatergic neurons release higher amounts of glutamate in the Glu-CB_1_R-KO mouse strain [41], and that glutamate enhances the proliferation of 1014 and other melanoma cell lines in vitro [45].

NMDARs are heterotetramers composed of two NMDAR1 subunits and two NMDAR2 or two NMDAR3 subunits, which are usually exclusive. Intriguingly, disrupting mutations in the *GRIN2A* gene are frequent in melanoma [58], and they are associated with bad prognosis [59]. To reconcile these apparently paradoxical observations, we speculate that the subunit composition of NMDARs, which dramatically impacts the receptor’s functional properties [60], might influence melanoma cell growth. Strikingly, in the GDC-TCGA SKCM dataset, a high expression of *GRIN3A* is associated with longer overall survival (low *GRIN3A* median survival: 1960 days; high *GRIN3A* median survival: 4930 days; *p* < 0.0001 by Log-rank (Mantel–Cox) test). Therefore, a plausible explanation to be explored in the future is that mutant forms of *GRIN2A* and/or a low expression of *GRIN3A* favour the integration of other subunits, such as NMDAR2B, that, in turn, alter NMDAR’s biophysical properties to favour a proliferative signalling axis. Further research is warranted as well to identify downstream effectors that, upon NMDAR activation, sustain the proliferation of MBMs.

On the other hand, the potential involvement of the ECS, and particularly of CB_1_Rs, in the growth of MBMs, should come as no surprise, as the activation of this receptor generally occurs when homeostasis is perturbed in the brain and other tissues [61]. Pharmacological activation of CB_1_Rs upon systemic administration of cannabinoid agonists triggers tumour-cell apoptosis in numerous mouse models of cancer, especially glioblastoma, the most common form of brain cancer [62,63], but to which extent this event relies, at least in part, on CB_1_R molecules located on tumour-microenvironment-forming cells (such as neurons in MBMs) remains largely unknown. It has been previously shown that cannabinoid administration inhibits melanoma cell growth in subcutaneous-allograft mouse models, and limits liver and lung colonisation by melanoma cells [10,12]. We are aware that one of the shortcomings of the present study is that no pharmacological experiments have been conducted on MBMs in vivo. A drug treatment, in principle, is not cell population-specific. In our case, broadly speaking, a systemic cannabinoid administration to mice bearing MBMs will target every body cell expressing CB_1_Rs or CB_2_Rs, therefore precluding the interpretation of the results obtained. Ideally, to dissect the precise subpopulations of CB_1_Rs involved, these experiments should be conducted with (i) different doses of a CB_1_R-selective antagonist and a CB_1_R-selective agonist (plus their vehicle control); (ii) various lines of age-matched, genetically-modified mice (at least Glu-CB_1_R-KO and GABA-CB_1_R-KO mice, plus their CB_1_R-floxed littermates); and (iii) CB_1_R-WT and CB_1_R-KO melanoma cells. These experiments are beyond the scope of the present study and will be the subject of future research by our group.

## 5. Conclusions

This work provides a novel conceptual framework to understand the molecular factors that contribute to MBMs. The demonstrated dependency of melanoma cell proliferation on NMDARs, and the key role of CB_1_Rs located on excitatory nerve terminals in the control of glutamate release, suggest that activating CB_1_Rs selectively on those terminals and/or inhibiting (brain) melanoma-specific NMDAR downstream effectors might be evaluated as novel therapeutic interventions to suppress MBMs. We therefore anticipate that this study may help to design strategies aimed at targeting neuronal glutamatergic signalling in MBMs.

## Data Availability

All source data generated or analysed during the study are included in this article.

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
