# Peer review of "Neuronal Cannabinoid CB1 Receptors Suppress the Growth of Melanoma Brain Metastases by Inhibiting Glutamatergic Signalling"

_cancers, 2023, doi:10.3390/cancers15092439_

Round 1
Reviewer 1 Report
In their work, Costas-Insua and colleagues described their observations on how cannabinoid receptors influence the growth of melanoma brain metastases. Although the topic is of great interest and the data is well presented, the authors conclusions should be considered with care and reinforced by additional experiments, for this manuscript to be eligible for publication in Cancers and useful to the community. Please find below a list of comments and suggestions for major revision of the work.
(1) Figure 1 and 2:
In spite of statistically significant differences between PT vs Metastasis samples, the extent of these different appear very modest, likely not so relevant on a biological point of view.
a. Why did the authors chose to normalize the data to 100%, instead of providing and comparing FKPM data as such ?
b. GDC-TCGA data is based on bulk RNA from heterogeneous tissue pieces. How to correct for possible neuronal/astrocytic contamination in Metastasis samples ?
c. Could the Authors access single-cell datasets that could confirm these gene up/down regulations in MDM cells exclusively ?
d. Is there any clear correlation between ECS genes, glutamate receptors and patient clinical outcome ?
(2) Table 2:
a. I suggest to present the qPCR data on melanoma cell lines as fold expression to several housekeeping genes.
b. Could the expression of the receptors at the cell membrane be confirmed by western-blot or flow cytometry ?
c. Do human melanoma cell lines harbor a similar expression of glutamate-related genes ?
(3) Figure 3:
a. The dose-response curves are compared to a vehicle condition that should be plotted as well.
b. Could these results be reinforced by observing the impact of glutamate deprivation? Or in rescue experiments with increased glutamate concentrations?
(4) Figure 4:
a. Although these mice constructs have been previously validated, did the authors verify the correct KO of CB1R in Glu or GABA neurons ?
b. To confirm MRI measurements, could tumor size be assessed by histology ?
c. What is the impact of CB1R activation by cannabinoids (THC, CBD,…) in WT mice with melanoma ? Does it limit MDM formation ?
(5) Figure 5:
a. How to correct for neuronal/astrocytic contamination in tumors (% purity) ? (for the PSD95 staining quantification but also certainly for the protein analysis after tissue dissection ?
b. pNR2B should be normalized to total NR2B prior to tubulin.
c. Is there a difference in the clinical outcome of KO vs WT mice ? (body weight, survival time,...)
(6) Figure 6:
a. Is MBM formation observed in the subcutaneous injection melanoma model ? This one would actually be most relevant.
b. similar to Fig5: Is there a difference in the clinical outcome of KO vs WT mice ? (body weight, survival time,...)
Author Response
REVIEWER 1
In their work, Costas-Insua and colleagues described their observations on how cannabinoid receptors influence the growth of melanoma brain metastases. Although the topic is of great interest and the data is well presented, the authors conclusions should be considered with care and reinforced by additional experiments, for this manuscript to be eligible for publication in Cancers and useful to the community. Please find below a list of comments and suggestions for major revision of the work.
We would like to thank the reviewer very much indeed for his/her positive and constructive comments, which we frankly believe have helped to improve the quality of our study.
(1) Figure 1 and 2:
In spite of statistically significant differences between PT vs Metastasis samples, the extent of these different appear very modest, likely not so relevant on a biological point of view.
The reviewer is correct. We have acknowledged in the text that the observed changes were modest/moderate (lines 353 and 371).
- Why did the authors chose to normalize the data to 100%, instead of providing and comparing FKPM data as such ?
We thank the reviewer for pointing this issue. We normalized the values of metastases to the mean expression in primary tumors to facilitate the comparison across genes to the readers, but the reviewer is correct in suggesting this alternative manner of representing the data as the absolute expression of the different genes was missing in our relative quantification. We now provide the FPKM data in Figs. 1 and 2 (see also their legends, lines 362 and 415).
- GDC-TCGA data is based on bulk RNA from heterogeneous tissue pieces. How to correct for possible neuronal/astrocytic contamination in Metastasis samples ?
We agree with the reviewer. We are aware of this potential limitation of this and any other study conducted with this type of samples. Unfortunately, single-cell RNAseq studies from primary melanoma and matched MBM samples are very scarce (see below), thus restricting our options for analysis. We frankly believe, however, that our in vitro and in vivo results provide enough evidence to support a pro-oncogenic role for the glutamate/NMDAR axis in MBM growth.
- Could the Authors access single-cell datasets that could confirm these gene up/down regulations in MDM cells exclusively ?
We thank the reviewer for this suggestion. Unfortunately, to the best of our knowledge, only one study has performed RNAseq on treatment-naïve MBMs (PMID: 35803246), and no matching primary tumors were included to allow a proper comparison. Nonetheless, the authors of that report stated that metastatic melanoma cells adopt a neuronal-like phenotype, which is consistent with our findings.
- Is there any clear correlation between ECS genes, glutamate receptors and patient clinical outcome ?
We thank the reviewer for this comment. As previously discussed in the manuscript (lines 569-577), patients with high GRIN3A expression show an increased overall survival compared to patients with low GRIN3A expression. This also applies when restricting the analysis to patients bearing metastatic melanoma. We have analyzed other markers relevant for our study and, aside from GRIN3A, only high CNR2 expression is associated with longer survival in the same cohort (p = 0.01705).
(2) Table 2:
- I suggest to present the qPCR data on melanoma cell lines as fold expression to several housekeeping genes.
We thank the reviewer for this proposal. Nonetheless, the data shown in Table 2 do not aim to make a comprehensive comparison between the different cell lines, but to support the presence of glutamate receptor subunits in the three of them based on a simple and widely used parameter as the Ct. For this reason, we would suggest that the data might stay as they stand.
- Could the expression of the receptors at the cell membrane be confirmed by western-blot or flow cytometry ?
We agree with the reviewer that this is a potential shortcoming of our study. Nonetheless, several studies have demonstrated a functional activity of NMDARs in melanoma cell lines (e.g., PMID: 27659111, 24739903, 23171453) as well as in normal melanocytes (PMID: 16420247). The dose-dependent effect of MK-801 observed in our study also points to the presence of active (thus, conceivably, plasma membrane-positioned) NMDARs. In sum, we frankly believe that all this information provides strong support to the presence of plasma membrane NMDARs in melanoma cells.
- Do human melanoma cell lines harbor a similar expression of glutamate-related genes ?
Yes, they do. We have analyzed RNAseq data from the Cancer Cell Line Encyclopedia. They clearly show a widespread expression of NMDAR subunits and related genes in a collection of human melanoma cell lines. These data have been included in the accompanying Figure for Reviewer 1.
(3) Figure 3:
- The dose-response curves are compared to a vehicle condition that should be plotted as well.
The experiment shown in Figure 3 represents dose-response curves, in which increasing concentrations of the drugs were applied. To do so, we added the same amount of vehicle to each well upon performing serial dilutions of the drugs. Thus, the vehicle condition (representing the mean of firstly technical, and secondly biological replicates) is a single, zero-drug-dose value, and thus cannot be represented as a curve. We apologize for this potential confusion, which has now been clarified in the Material and Methods section (lines 163-167).
- Could these results be reinforced by observing the impact of glutamate deprivation? Or in rescue experiments with increased glutamate concentrations?
We agree with the reviewer that this could be an interesting experiment. However, most commercially available cell-culture media contain glutamate and other amino acids that can be used by cells to synthesize glutamate, as well as co-activators of NMDARs such as glycine and D-serine. Nevertheless, in order to circumvent this problem, we made several attempts to grow the cells in chemically-defined media lacking amino acids, but cells were unable to proliferate properly in our hands under these conditions. These issues notwithstanding, we frankly believe that our cell-culture experiments, together with the accompanying in vivo assays, provide strong evidence for the involvement of the glutamate/NMDAR axis in MBM growth.
(4) Figure 4:
- Although these mice constructs have been previously validated, did the authors verify the correct KO of CB1R in Glu or GABA neurons ?
As the reviewer acknowledges, these genetically-modified mouse lines are widely validated for about 15 years by many groups in the field, including ours. In this study, as in others, we did indeed genotype all and every animal of the colonies to assess the absence/presence of Cre recombinase.
- To confirm MRI measurements, could tumor size be assessed by histology ?
This is a good idea. However, on the one hand, our group has long experience in state-of-the-art procedures of mouse-brain MRI techniques, so we are confident that they provide us with highly reliable data. On the other hand, the perfused-brain samples we obtained have been largely used for confocal microscopy determinations, and conducting brain-stereology experiments with new age-matched animals of the different conditional-knockout lines would be an extremely hard task that is beyond this study.
- What is the impact of CB1R activation by cannabinoids (THC, CBD,…) in WT mice with melanoma ? Does it limit MDM formation?
We and others have previously shown that cannabinoids inhibit melanoma cell growth in subcutaneous-allograft mouse models (see refs. 10 and 12 in the manuscript). The experiments proposed by the reviewer are no doubt interesting but, honestly, fall beyond the scope of the present study. A drug treatment, in principle, is not cell population-specific. Broadly speaking, a systemic cannabinoid administration will target every body cell expressing CB1R or CB2R receptors, therefore precluding the interpretation of the results obtained. Ideally, to dissect the precise subpopulations of CB1R involved, these experiments should be conducted with (a) different doses of a CB1R-selective antagonist and a CB1R-selective agonist (plus their vehicle control), (b) various lines of age-matched, genetically-modified mice (at least, Glu-CB1R-KO and GABA-CB1R-KO mice, plus their CB1R-floxed littermates), and (c) CB1R-WT and CB1R-KO melanoma cells. Hence, this would constitute an extremely human/logistics/time-demanding task. We have nonetheless acknowledged explicitly in the manuscript this limitation of the study and those challenging ideas for future research (lines 585-598).
(5) Figure 5:
- How to correct for neuronal/astrocytic contamination in tumors (% purity) ? (for the PSD95 staining quantification but also certainly for the protein analysis after tissue dissection ?
We agree with the reviewer that this is a potential limitation of this and any other study conducted with this type of samples. Nonetheless, we were always extremely careful in studying brain samples with the highest enrichment in melanoma cells and the lowest enrichment in neurons/astrocytes, so we frankly believe that neurons and other CNS-resident cell represent a very minor component of our tumor samples. In the case of confocal microscopy studies, non-tumor cells, mostly neurons, were easily differentiated both by the size of their nuclei and by the pattern of PSD95 staining, which is readily evident in the surrounding tumor area, but not within the tumor core. In addition, the injected melanoma cells contain melanin, and thus we defined the (tumor) quantifiable black area by acquiring images under bright field. A similar procedure applied to the Western blot assays, and so only black tissue was dissected under a magnifying glass. Since the amount of protein required to detect pNR2B was rather high (see below), we honestly believe that essentially all the observed signal originated from melanoma cells.
- pNR2B should be normalized to total NR2B prior to tubulin.
The reviewer in correct. In fact, we reblotted the pNR2B membrane with several commercial anti-NR2B antibodies but, unfortunately, we failed to obtain a reliable staining. Moreover, all these attempts required high cell protein amounts (up to 40 micrograms per sample) for detection, and thus we only had one trial per mouse/antibody. Hence, we were left with the only choice of normalizing to a standard housekeeping gene, and so used tubulin as a loading control, which we frankly believe is also a reliable assessment.
- Is there a difference in the clinical outcome of KO vs WT mice? (body weight, survival time,...)
This is a good remark. Regarding body weight, we did not find any significant difference between mouse genotypes in this intracranial experimental setting. Regarding survival, the Ethical Committee of our institution would not permit conducting that experiment.
(6) Figure 6:
- Is MBM formation observed in the subcutaneous injection melanoma model ? This one would actually be most relevant.
We did not observe any MBM formation in the subcutaneous melanoma model. We agree that it would have been certainly a very relevant metastasis model.
- similar to Fig5: Is there a difference in the clinical outcome of KO vs WT mice ? (body weight, survival time,...)
Again, we did not find any significant difference in body weight between KO and WT animals in any of the three experimental settings (namely, intracarotid, subcutaneous, and tail-vein melanoma cell injection). Regarding survival, the experiment is precluded by the Ethical Committee of our institution.

Reviewer 2 Report
Costas-Ensua et al hypothesize that cannabinoid C1 receptors located on glautamatergic nerve term goals may control the growth of melanoma brain metastases (MBMs).
Simple summary and abstract: Both describe the hypothesis and, in the abstract, briefly describes the study methods. Both describe the outcome. Writing is concise and consistent with the fax provided in the overall draft.
Introduction: Adequately describes the hypothesis, including the previous data providing a basis for the hypothesis and experimental design. I would recommend one change. On line 91, the word “hijack” needs a better definition. Please define this word or use different words to convey what the melanoma brain metastases specifically do to the astrocytes.
Materials and methods: Adequately describes the methodology of the experiments. There are no comments pertaining to the experimental design and associated procedures.
Results: Concisely describes data obtained from the described experiments, and results are easy to understand.
Figures and tables: All are appropriate (figures 1-6 including figures 5B and 5C and tables 1 and 2), and are not redundant with regard to the narrative description of the data, and enhance understanding of the data. Figures 5B and 5C were provided in a separate file for review, and I would like to know where they will be placed in the text.
Discussion: The discussion is concise and adequately interprets the results and potential future translation potential.
Conclusions: Appropriately brief, adequately describing the findings have potential for future study
References: Appropriate, no issues found
Author Response
REVIEWER 2
Costas-Ensua et al hypothesize that cannabinoid C1 receptors located on glautamatergic nerve term goals may control the growth of melanoma brain metastases (MBMs).
Simple summary and abstract: Both describe the hypothesis and, in the abstract, briefly describes the study methods. Both describe the outcome. Writing is concise and consistent with the fax provided in the overall draft.
Introduction: Adequately describes the hypothesis, including the previous data providing a basis for the hypothesis and experimental design. I would recommend one change. On line 91, the word “hijack” needs a better definition. Please define this word or use different words to convey what the melanoma brain metastases specifically do to the astrocytes.
Materials and methods: Adequately describes the methodology of the experiments. There are no comments pertaining to the experimental design and associated procedures.
Results: Concisely describes data obtained from the described experiments, and results are easy to understand.
Figures and tables: All are appropriate (figures 1-6 including figures 5B and 5C and tables 1 and 2), and are not redundant with regard to the narrative description of the data, and enhance understanding of the data. Figures 5B and 5C were provided in a separate file for review, and I would like to know where they will be placed in the text.
Discussion: The discussion is concise and adequately interprets the results and potential future translation potential.
Conclusions: Appropriately brief, adequately describing the findings have potential for future study
References: Appropriate, no issues found
We would like to thank the reviewer very much indeed for his/her positive and constructive comments on our study. Regarding the Introduction, as indicated, we have changed the term “hijack” (lines 105-107). Regarding the figures, following the editorial guidelines of the journal, we provided a supplementary file with the uncropped Western blots shown in Fig. 5B (Figure S1). Please note that there is no Fig. 5C in the paper. We are sorry if these issues were not correctly explained. Now we have cited explicitly the Supplementary Materials in the text (lines 306-307 and 608).
Reviewer 3 Report
General comments:
In this manuscript, the authors investigate the role of the ECS system in melanoma brain metastases. They show increased expression of ECS elements, including CB1R in metastatic versus primary melanoma using public datasets. They also show that blockade of NMDA receptors reduces growth in melanoma cell lines. Finally, they show that loss of CB1R is associated with increased growth of melanoma brain metastasis models, but is not associated with increased extravasation of metastatic tumor cells. This study seeks to answer a key mechanistic question about a major issue in cancer care. Overall, the study is well-designed, with good integration of complementary in silico, in vitro, and in vivo data. The data is well presented. Some additional points of clarification are needed. I have also suggested some additional experiments which the authors should at least consider, if they are not beyond the scope of the study.
Specific comments:
1) Introduction, line 69: it may be helpful to illustrate the ECS
2) Introduction, line 79: what other types of cancer cells? Please list them specifically.
3) Methods 2.10 and 2.12: These experiments are very interesting conceptually, but have they been previously validated as reliable assays of metastatic tumor cell extravasation? No papers are cited.
4) Results 3.1: were the metastatic melanoma samples brain metastases specifically?
5) Results 3.3, line 435: Would it make sense to measure CNS glutamate levels at the time the brain is harvested? I know it has been shown in previous studies of these knockouts, but showing a direct relationship between absence of CB1R, increased glutamate, increased p-NMDAR2B, and increased tumor growth all in the same experiment would lend even stronger support to the hypothesis, would it not?
6) Results 3.3, line 480: What is the relative expression of CB1R in the brain, lungs, and skin?
7) Discussion: the data immediately suggest 2 experiments in genetically normal mice: 1) inhibiting CB1R and looking for tumor growth, and 2) stimulating CB1R and looking for tumor regression. The authors allude to the second experiment being done in “numerous mouse models of cancer,” but has it been done with metastatic melanoma? Is this type of experimentation beyond the scope of the authors’ study? The results would be interesting to see.
Author Response
REVIEWER 3
In this manuscript, the authors investigate the role of the ECS system in melanoma brain metastases. They show increased expression of ECS elements, including CB1R in metastatic versus primary melanoma using public datasets. They also show that blockade of NMDA receptors reduces growth in melanoma cell lines. Finally, they show that loss of CB1R is associated with increased growth of melanoma brain metastasis models, but is not associated with increased extravasation of metastatic tumor cells. This study seeks to answer a key mechanistic question about a major issue in cancer care. Overall, the study is well-designed, with good integration of complementary in silico, in vitro, and in vivo data. The data is well presented. Some additional points of clarification are needed. I have also suggested some additional experiments which the authors should at least consider, if they are not beyond the scope of the study.
We would like to thank the reviewer very much indeed for his/her positive and constructive comments, which we frankly believe have helped to improve the quality of our study.
Specific comments:
1) Introduction, line 69: it may be helpful to illustrate the ECS
Good idea. This could help the general readership of the study. So, we have included a scheme of the ECS in the paper (Scheme 1, lines 78-87).
2) Introduction, line 79: what other types of cancer cells? Please list them specifically.
There is a very large number of cancer cells in which the antitumoral activity of cannabinoids has been shown. As indicated by the reviewer, we have mentioned now in the text the most thoroughly studied of them (lines 89-93).
3) Methods 2.10 and 2.12: These experiments are very interesting conceptually, but have they been previously validated as reliable assays of metastatic tumor cell extravasation? No papers are cited.
Yes, these experimental settings are well validated in the field to assess the metastatic potential of cancer cell lines (see, for example, PMID: 18228345, 30679540, 31534217, 35993751). The carotid-artery injection procedure was previously referenced in the text (ref. 44, line 492), and now we have also referenced the tail-vein injection procedure (ref. 20, line 498).
4) Results 3.1: were the metastatic melanoma samples brain metastases specifically?
Good point indeed. Metastatic melanoma samples had different origins, mostly unannotated (81.7%). In the annotated samples (19.3% of the total), 1.6% of the total were from brain origin, so too few to conduct a robust statistical analysis. Thus, assuming a normal distribution, we estimate that 31 out of 368 samples might be MBMs. Since primary tumor/MBM-matched samples are currently unavailable, we speculate that our observed changes are indeed greater that observed as they may have become “diluted” in the dataset.
5) Results 3.3, line 435: Would it make sense to measure CNS glutamate levels at the time the brain is harvested? I know it has been shown in previous studies of these knockouts, but showing a direct relationship between absence of CB1R, increased glutamate, increased p-NMDAR2B, and increased tumor growth all in the same experiment would lend even stronger support to the hypothesis, would it not?
The reviewer is correct. This would be a nice experiment. Nonetheless, our group, using microdialysis techniques, has previously reported that the concentration of extracellular glutamate is increased in the dorsal striatum (i.e., precisely where we injected the melanoma cells in this study) of Glu-CB1R-KO mice (see ref. 41, lines 447 and 562 in the manuscript). On top of that, these Glu-CB1R-KO mice have been used by different groups -including ours- for about 15 years (their generation dates back to 2006, see ref. 27 in the manuscript) to demonstrate, from a functional standpoint, that CB1R indeed blocks glutamatergic signaling in different regions of the brain, including the dorsal striatum. Hence, based on these previous findings, together with the extreme logistical and technical complexity of the experiments proposed by the reviewer, we frankly believe that the present study provides strong in vivo evidence for the role of CB1R in the control of the glutamate/NMDAR/MBM proliferation axis.
6) Results 3.3, line 480: What is the relative expression of CB1R in the brain, lungs, and skin?
CB1R is highly expressed in the human brain and is remarkably less abundant in the lungs and the skin. See the accompanying Figure for Reviewer 3 (data taken from GTEx portal). A similar expression pattern occurs in mouse; see, for example:
https://www.ebi.ac.uk/gxa/genes/ensmusg00000044288?bs=%7B%22mus%20musculus%22%3A%5B%22ORGANISM_PART%22%5D%7D&ds=%7B%22kingdom%22%3A%5B%22animals%22%5D%7D#baseline
https://www.informatics.jax.org/marker/MGI:104615
7) Discussion: the data immediately suggest 2 experiments in genetically normal mice: 1) inhibiting CB1R and looking for tumor growth, and 2) stimulating CB1R and looking for tumor regression. The authors allude to the second experiment being done in “numerous mouse models of cancer,” but has it been done with metastatic melanoma? Is this type of experimentation beyond the scope of the authors’ study? The results would be interesting to see.
We agree with the reviewer, though, as he/she asks, we honestly believe that these interesting experiments fall beyond the scope of the present study. A drug treatment, in principle, is not cell population-specific. Broadly speaking, a systemic cannabinoid administration will target every body cell expressing CB1R or CB2R, therefore precluding the interpretation of the results obtained. Ideally, to dissect the precise subpopulations of CB1R involved, these experiments should be conducted with (a) different doses of a CB1R-selective antagonist and a CB1R-selective agonist (plus their vehicle control), (b) various lines of age-matched, genetically-modified mice (at least, Glu-CB1R-KO and GABA-CB1R-KO mice, plus their CB1R-floxed littermates), and (c) CB1R-WT and CB1R-KO melanoma cells. Hence, this would constitute an extremely human/logistics/time-demanding task. We have nonetheless acknowledged explicitly in the manuscript this shortcoming of our study and those challenging ideas for potential future research (lines 585-598). Regarding the other question posed by the reviewer, previous work (refs. 10 and 12 in the manuscript, lines 585-587) shows that administration of cannabinoids to mice, aside from inhibiting subcutaneous melanoma cell growth, limits liver and lung colonization by melanoma cells.

Round 2
Reviewer 1 Report
The Authors addressed most of the Reviewer's comments. They implemented some more information in their manuscript, however I think more content from their replies is worth being accessible to the readers (at least via the availability of the review report).